# Golgi Fragmentation in Neurodegenerative Diseases: Is There a Common Cause?

**DOI:** 10.3390/cells8070748

**Published:** 2019-07-19

**Authors:** José Ángel Martínez-Menárguez, Mónica Tomás, Narcisa Martínez-Martínez, Emma Martínez-Alonso

**Affiliations:** 1Department of Cell Biology and Histology, Medical School, Biomedical Research Institute of Murcia (IMIB-Arrixaca-UMU), University of Murcia, 30100 Murcia, Spain; 2Department of Human Anatomy and Embriology, Medical School, Universitat de Valencia, 46010 Valencia, Spain

**Keywords:** Golgi complex, neurodegenerative diseases, intracellular transport, cytoskeleton, protein aggregates

## Abstract

In most mammalian cells, the Golgi complex forms a continuous ribbon. In neurodegenerative diseases, the Golgi ribbon of a specific group of neurons is typically broken into isolated elements, a very early event which happens before clinical and other pathological symptoms become evident. It is not known whether this phenomenon is caused by mechanisms associated with cell death or if, conversely, it triggers apoptosis. When the phenomenon was studied in diseases such as Parkinson’s and Alzheimer’s or amyotrophic lateral sclerosis, it was attributed to a variety of causes, including the presence of cytoplasmatic protein aggregates, malfunctioning of intracellular traffic and/or alterations in the cytoskeleton. In the present review, we summarize the current findings related to these and other neurodegenerative diseases and try to search for clues on putative common causes.

## 1. Introduction

Neurodegenerative diseases share many aspects in their late phases, and it is possible that they are originated by common pathological mechanisms. In all of them, for example, there is an extra- and intracellular accumulation of abnormal proteins, the progressive failure of basic cellular processes that results in synaptic alterations and, finally, these catastrophic events end in neuronal death [1]. The origins of these degenerative processes are not well known but seem to be triggered by multiple intrinsic and extrinsic factors. Each disease affects a selective group of neurons and when the damage is considerable, the specific symptoms of each disease begin to manifest themselves. The cellular processes affected include protein dynamics, bioenergetics and mitochondrial function, oxidative balance, axonal transport, etc. [2]. Genes involved in genetic forms of neurodegenerative diseases are not exclusive to the affected cells. In fact, they are related to common cellular processes and expressed in a wide range of cell types; however, it is unclear why just a limited number of neurons are affected. One of the common features of neurons during neurodegeneration is alteration of the organization of the Golgi complex (GC). In neurons and in the vast majority of mammalian cells, this organelle forms a continuous ribbon. During the progress of the disease, this organization is broken into small isolated elements [3,4]. Importantly, Golgi fragmentation is an early preclinical event in many neurodegenerative diseases and precedes other alterations. Golgi fragmentation is not a consequence of senescence [5] and has been observed in neurons affected by pathologies such as hydrocephalus [6] and Dyggve–Melchior–Claussen syndrome, a type of skeletal dysplasia which is included in the group of Golgipathy disorders [7]. In this review, we first provide an overview of the organization of GC in neurons and how it is affected in neurodegenerative disease. We then analyze the putative causes of the fragmentation of the Golgi ribbon in different diseases and, finally, we try to find a common cause.

## 2. The Golgi Ribbon

The GC is typically formed of 3–9 cisternae arranged very closely to each other and aligned in a parallel way, similar to a pile of coins, forming the so-called Golgi stacks or dictyosomes [8,9]. Different types of vesicles, tubules and, in specialized cells, secretory granules surround the stacks. In most mammalian cells, the stacks are laterally connected by tubules forming a continuous ribbon. This ribbon organization is clearly observed in 3D studies with electron tomography [10]. The GC is the central station of the secretory pathway. Newly synthesized proteins and lipids exit the endoplasmic reticulum (ER) via COPII-coated vesicles, arrive at the endoplasmic reticulum–Golgi intermediate compartment (ERGIC) and then the cis Golgi cisternae cross the stack by means of a not well understood mechanism and finally, they are packed into the trans-Golgi network (TGN) and delivered to their final destination. In addition, there is a COPI-mediated Golgi-to-ER retrograde transport, and the GC/TGN also receives materials from the endocytic pathway. Transport between compartments is mediated by coated vesicles (and also tubules and other heterogeneous transport intermediates). The molecular machinery involved in the formation, transport, docking and fusion with the donor membranes is well known and includes coat complexes, tethering factors, Rab GTPases and v (vesicular)- and t (target)-SNAREs [11]. The balance between vesicle formation at the donor membranes and their docking and fusion with the target membranes is important for the maintenance of the structure, as can been observed during mitosis.

Briefly, Golgi stack and ribbon organizations are maintained by a complex machinery involving (1) Golgi matrix proteins, including GRASP55, GASP65, GM130, Golgin-45, Golgin-84 and Golgin-160; (2) proteins of the transport machinery (tethering proteins and complexes, members of the Rab family of small GTPases, Golgi-associated SNARE proteins); (3) microtubules and their associated motor proteins and proteins involved in the interaction of this organelle with the cytoskeleton; (4) signaling proteins; and (5) proteins related with pH and calcium homeostasis [8,12,13]. Deletion or over-expression of these proteins alters their organization significantly and importantly, some of them are involved in physiological and pathological modifications of Golgi structure.

Microtubules are also important for Golgi organization and positioning. In fact, the GC is an important place for the nucleation of microtubules and Golgi membranes have the machinery necessary for the nucleation, regulation and stabilization of microtubules, acting as a secondary microtubule organizing center (MTOC) [14]. The microtubule network is important for pre- and post-Golgi transport and for the maintenance of cell polarity. Microtubules are especially important in neurons. The high rate of transport of organelles, proteins and RNA granules along the long axons, the development of neuronal polarity and neuronal migration and polarization during brain morphogenesis depend on microtubules. The GC is an essential microtubule-organizing center in neurons, given that centrosome loses its function during differentiation [15]. Golgi-related organelles (see below) could act as nucleation sites for dendritic microtubules. The ribbon-like organization of the GC depends on microtubules. It is well known that microtubule de-polymerization with the drug nocodazole induces the fragmentation of the Golgi ribbon in mini-stacks, which are redistributed throughout the cytoplasm. These mini-stacks are functional in terms of transport rate (except for large cargo) or glycosylation [16]. Nocodazole-induced mini-stacks do not redistribute randomly; in fact, they move to the ER-exit sites [17], which explains why the traffic is not altered. It is not known whether this is the case of Golgi fragmentation in neurodegenerative diseases. Importantly, the polarized transport is lost in nocodazole-treated cells [16]. Thus, the intact Golgi ribbon could be important for maintaining the polarization of membranes. If this loss of polarization indeed happens in neurons affected by neurodegenerative diseases, it may have dramatic effects on their physiology. The Golgi ribbon also depends on microtubule motors, especially dynein, whose recruitment to Golgi membranes is mediated by Golgin160 in a process which is regulated by Arf1 [18]. The main role of the small GTPase ARF1 is the recruitment of coatomer (a cytosolic complex formed by COPI subunits) to membranes [19]. Thus, there is a link between dynein recruitment and COPI formation. There is also a link between COPI and tubulin polymerization at the GC. ARF1 also enhances the recruitment at the Golgi membranes of the Golgi-localized tubulin-binding cofactor E (TBCE), which is necessary for microtubule polymerization [20].

Actin microfilaments are also necessary for the structure of the GC and a number of actin-binding proteins have been located in its membranes, such as Arp2/3 complex, myosins and spectrin [21,22,23]. In contrast to microtubule depolymerization, actin-disrupting agents induce the compaction of the GC [24]. How actin-based cytoskeleton contributes to the maintenance of the ribbon is not well known. A recent study suggested that the ribbon structure is regulated by the actin cytoskeleton and myosin IIA by interacting with GCC88, a TGN golgin [25].

This ribbon can be dissembled under physiological conditions such as mitosis [26,27,28]. During the G2 phase of the cell cycle, lateral unlinking of the stacks is mediated by the stacking proteins GRASP65 and GRASP55 and the fission-inducing protein BARS, which break the tubular connections between stacks. Phosphorylation of GRASP65 by the JNK2 and other kinases induces the disruption of the ribbon and unstacking cisternae [28]. Phosphorylation of GRASP55 by ERK2 also induces ribbon unlinking. In prophase, the stacks are separated into individual cisternae, a process regulated by phosphorylation of those stacking proteins by CDK1. Finally, the cisternae are further fragmented into vesicles. Vesiculation occurs due to the fact that vesicle formation from cisternae continues but fusion is inhibited. In the telophase, the Golgi ribbon is reassembled in daughter cells.

Golgi disassembly during apoptotic cell death occurs in a similar manner [29]. Fragmentation is an early event that coincides with the release of cytochrome c from mitochondria prior to alteration of the actin and microtubule cytoskeleton [30]. As described for mitosis, during apoptosis, the GC is dissembled in mini-stacks and then vesiculation occurs [29,31]. Fragmentation is an early event independent of the cytoskeleton [30], whereby secretion is blocked by inhibition of the ER exits [32]. The Golgi is a sensor of mitochondrial and ER stress [33]. Fragmentation of the GC may activate a stress signaling pathway and, if this stress is not corrected, the result is cell death [34]. Unlinking Golgi stacks or the alteration of vesicular trafficking might lead to the induction of the death receptor DR4 and its accumulation in the Golgi, activating the Golgi stress response, analogous to the unfolded protein response (UPR) of the ER [35]. Conversely, fragmentation could be part of the irreversible phenomenon that occurs during cell death. Golgi structural proteins are cleaved by caspases during apoptosis. The golgin tether giantin, the t-SNARE syntaxin 5 [32] and GRASP-65 [36] are selectively cleaved by caspase 3. Caspase 2 has been located in the Golgi membranes [37]. This initiator caspase acts earlier than caspase 3 and may play an early role in GC activation. Caspase regulators, anti-apoptotic proteins and death receptors have also been detected in the GC [34].

Why is the ribbon necessary? It is clear that the ribbon is not necessary for basic functions of the GC such as glycosylation and sorting, since this organization is not observed in non-vertebrate cells or even in some mammalian cells such as oocytes and skeletal muscle cells [38]. Fragmentation does not affect transport although it accelerates the movement of large cargo [39]. The fragmented GA remains functional [16,40]. The ribbon has been implicated in “high-order” functions such as the regulation of the mitosis entry (Golgi checkpoint), apoptosis, stress response (Golgi stress response), establishment and maintenance of cell polarity, directional cell migration, packaging and regulation of the size of special secretory granules and the regulation of the distribution of glycosylation enzymes between stacks [13,41]. The Golgi ribbon also negatively regulates autophagy [42]. The importance of GC is remarked by the fact that 159 signaling genes regulate the morphology of this organelle and the deletion of 55 of them induces Golgi fragmentation [43].

## 3. The Golgi Complex of Neurons

The neuron was the first cell type where the GC was described more than 120 years ago by Camillo Golgi. In neurons, GC is very well developed, forming a ribbon that surrounds the nucleus, as demonstrated by the use of cytochemical techniques [44,45]. The importance of the GC in neurons is underlined by the fact that the central nervous system is the organ/system most affected by Mendelian disorders caused by mutations in GC components [46]. In addition to the basic and high-order functions necessary in all cells, the GC of neurons is necessary for determining and maintaining axodendritic polarity [47]. The position of the GC is important for migration during brain development but also for dendrite formation in new neurons in the adult hippocampus [48]. The GC of neurons also has an important role in autophagy. Indeed, the GC itself can sequester the cytoplasmic contents, working simultaneously with autophagosomes in neurons [49].

Apart from the central or somatic GC, smaller related structures have been found in dendrites, the so-called Golgi outposts and satellites. Golgi outposts are scarce and restricted to the main branches of the dendritic arbor, although they are not present in every neuron [47]. These tubule-vesicular structures derived from the perinuclear GC [50] contain matrix proteins and glycosylation enzymes. They have been seen to be involved in the sorting, trafficking and posttranslational modification of post-Golgi carriers [51], as well as the main nucleation centers for microtubules in dendrites [52,53]. These activities are necessary for the growth and maintenance of the dendritic arbor. Although they contain cis, medial and trans Golgi compartments, they are usually disconnected [54]. Golgi satellites are more frequent than outposts and have been found alongside dendrites. They contain the glycosylation machinery but not the typical components of GC sorting and organization. They could be involved in the transport and recycling of some specific membrane proteins [55]. These data support the view that dendrites have specific machinery for the local delivery of specific membrane proteins such as synaptic receptors. Although cis, medial and trans Golgi markers have been identified in axons, a typical stacked GC has not been described [56]. ER-to-Golgi transport is necessary for dendritic growth but not for the axon in Drosophila neurons [57].

## 4. Fragmentation of the Golgi Ribbon in Degenerative Neurons

By light microscopy, the GC of neuron cells appears as dots or short filaments mainly distributed around the nucleus. In many neurodegenerative diseases, these dots become smaller and more widely distributed, a phenomenon denominated Golgi fragmentation [3,4]. However, electron microscopy is needed to determine whether, besides the lateral unlinking of the stacks, there is also unstacking of cisternae and vesiculation. Unfortunately, studies of the ultrastructure of the GC of neurons from human patients are scarce. Purkinje cells of AD patients showed some damage with cisternae slightly dilated and shorter, and fewer vesicles associated to them [58]. However, it must be taken into consideration that these cells did not have tau pathology and showed minimal amyloid deposits.

Most previous studies have been made in animals and cellular models of neurodegenerative diseases. Motor neurons from mice bearing a mutation in the TBCE gene, which encodes one of at least five tubulin specific chaperons used as a model of progressive motor neuronopathy, showed partially or completely vesiculated GCs [20]. Vesiculation increased with time. However, the GC of a cellular model of Parkinson’s disease did not show vesiculation, only ribbon unlinking [59]. The GC of Hippocampus and cortex neurons from APP Swedish mutation and exon deletion mutant of human PS1 transgenic mice (as a model of Alzheimer’s disease) showed swollen cisternae and disorganized stacks [60]. When CHO cells were used, instead of transgenic mice, the cisternae were seen to be shorter, fewer per stack and there were slightly more Golgi-associated vesicles, indicating some degree of unstacking and vesiculation.

It is evident that more ultrastructural studies of human samples are needed in order to determine how each neurodegenerative disease affects the lateral linking of stacks, cisternae stacking and the formation/consumption of Golgi vesicles. Importantly, each model used to study Golgi fragmentation must be checked to confirm that it properly mimics the real effects of the disease.

## 5. Putative Causes of Golgi Fragmentation in Neurodegenerative Diseases

The causes of GC fragmentation of the GC seem to be diverse. As indicated above, the maintenance of cisternae stacking and lateral linking depends on many structural and regulatory proteins. The up- or down-regulation or the cleavage by caspases of one single protein may have dramatic effects. Thus, for instance, deletion of matrix protein GM130 causes fragmentation, alteration of Golgi positioning, and impaired secretory traffic, which induces neuronal degeneration in KO mice [61].

As indicate above, alteration of the microtubule cytoskeleton (or even actin microfilaments and neurofilaments) may also be the cause. The depolymerization of microtubules by the drug nocodazole breaks the continuous ribbon into disconnected and dispersed mini-stacks resembling the GC of neurons in neurodegenerative diseases. During neurodegeneration, in addition to fragmentation, the isolated GCs are distributed throughout the cytoplasm [3,4], suggesting that the cytoskeleton is damaged or, at least, rearranged. In contrast with many other cell types, the distance between the ER exit sites, the GC and the plasma membrane can be very large in neurons [51]. Thus, even a small delay or inefficiency can have dramatic effects in neuronal function.

Neurons from neurodegenerative disease patients are characterized by the presence of intracellular and extracellular inclusions which may affect Golgi organization. Cytoplasmic inclusions can damage the GC simply by exerting mechanical forces that break the tubular connections between stacks. Moreover, cytoplasmic aggregation may sequester Golgi structural and regulatory proteins, preventing them from carrying out their normal function. Large inclusions may also alter the position of the GC within the cell.

An imbalance between incoming and out-going transport at the cis and trans/TGN sides may also cause fragmentation. Moreover, the disequilibrium between vesicle budding and docking/fusion may result in a massive vesiculation. Thus, a failure of the machinery in these processes may be responsible for fragmentation. Endocytic/endolysosomal transport is also altered in neurodegenerative diseases. Retrograde transport from early endosomes to the TGN is mediated by the membrane complex called retromer [62,63]. A dysfunctional retromer-mediate endosome-to-TGN trafficking may alter Golgi organization. The damage may become more pronounced because retromer recruits the molecular machinery (the WASH complex) necessary for actin nucleation and the formation of filaments necessary for the docking of endosomal vesicles [64].

Reversible Golgi fragmentation can be a consequence of hyperexcitability and increased neuronal activity, which is a common characteristic of neurodegenerative disease [65]. This fragmentation may alter calcium homeostasis, which is important for many neuronal activities.

Some of these causes have been investigated in several models of neurodegenerative diseases, the main conclusions of which are summarized below. Detailed descriptions for each disease can be seen in the excellent reviews quoted in the text.

## 6. Parkinson’s Disease (PD)

The GC (or more specifically the TGN) from PD patients appears fragmented in nigral neurons and is not related to cell death [66]. The main histopathological sign of the PD are the protein aggregates known as Lewy bodies. More than 70 proteins have been identified in these fibrillar aggregates [67]. α-Synuclein is the most abundant and appears phosphorylated and ubiquitinated, forming filamentous aggregates. The role of this protein is not known, but it has been postulated that it regulates the formation of synaptic vesicles from early endosomes by interacting with phospholipase D2. Another role of this protein may be the regulation of assembly of the soluble NSF attachment protein receptor (SNARE) complex, acting as a chaperone during synaptic processes and regulation of the synthesis and release of dopamine [68]. More recently, it has been shown that monomeric α-synuclein plays a role in sensing and generating curvature. Over-expression impairs tubulation and may be a mechanism involved in Golgi fragmentation [69]. The accumulation of α-synuclein, by failure of their degradation (autophagy, proteasome, lysosomal), altered expression (duplication/triplication) or expression of mutated forms, induces its aggregation [70,71,72]. Oligomers of α-synuclein, which are more dangerous than fibrils, can damage membranes by forming transmembrane pores [73]. Fragmented GC has been described in neurons with Lewy bodies from PD patients [66] and cells with α-synuclein-positive inclusions from patients with multiple system atrophy [74]. Interestingly, the percentage of cells with fragmented GC is similar (and very low) in neurons with Lewy bodies and neurons without inclusions in PD samples [66]. However, this number increases to one fifth of the cells with pale bodies, the precursors of Lewy bodies. Thus, it seems that inclusions repair rather than damage the GC. If this is true, it means that fragmentation is reversible and protein aggregates might sequester pathogenic factors such as over-expressed Golgi proteins or altered cytoskeleton components. For experiments in cell models confirm that Golgi fragmentation is an early event that occurs before the formation of fibrillar inclusions [59]. Conversely, this damage is caused by non-fibrillar oligomeric species of α-synuclein [75].

Most PD cases are sporadic with unknown etiology but 15% are familiar. At least 23 loci and 19 disease-causing genes for Parkinsonism, including α-synuclein, have been identified [76]. Mutations in LRRK2 (leucine-rich repeat kinase 2) are the most common genetic cause of PD and are also one of strongest risk factors for sporadic PD [76]. This kinase regulates internalization and the Golgi-to-plasma membrane trafficking of dopamine receptors [77]. It interacts with many partners, including members of the Rab GTPases family. One of these Rabs, Rab7L1, is associated with the GC. Pathogenic mutants of this kinase enhance the phosphorylation of Rab7L1 and alter their distribution and TGN morphology [78]. Other substrates of this kinase are Rab8 and Rab10. The Golgi location and activation of LRRK2 are mediated by Rab32, which enhances the phosphorylation of Rab 8 and 10 [79]. Interestingly, human LRRK2 and LRRK, the Drosophila melanogaster homologue, regulate the dynamics of Golgi outposts in dendrites [80] by inhibiting dynein-mediated movements. PD mutants of LRRK2 enhance the retrograde movement of Golgi outposts to the cell body. Another gene related with Parkinsonism is the phospholipase A2 group VI gene (PLA2G6). This enzyme catalyzes phospholipid and hydrolyzes glycerophospholipids to produce free fatty acid and lysophospholipid. In fibroblasts from patients carrying this mutation, N- and O-linked glycosylation profiles were altered and ERGIC and Golgi were damaged, which may contribute to neurodegeneration [81].

A milestone in the field was the discovery that, in yeast models, the major effect of α-synuclein toxicity was the inhibition of ER-to-Golgi vesicular transport, a block that can be reversed by the GTPase Rab1 [82]. This finding was confirmed in mammalian cells. Thus, the over-expression of wild type or disease-associated mutant α-synuclein delayed ER-Golgi transport by inhibiting vesicle docking and fusion [83]. Apart from this GTPase, Rab 8A and 3A reduce α-synuclein toxicity, indicating that post-Golgi trafficking is also affected [84]. Moreover, several Rabs have been involved in the regulation of aggregation, cell-to-cell propagation, recycling and inclusion clearance of α-synuclein [85]. In fact, this protein directly interacts with several Rabs, three of which (Rab3, Rab6, Rab8) have been related to ribbon organization [86]. The imbalance of the traffic of vesicles entering and leaving the GC could be the cause of the Golgi fragmentation [87]. Overexpression of α-synuclein also induces Golgi fragmentation in astrocytes [88]. A previous study by our group confirmed that Golgi fragmentation precedes the aggregation of α-synuclein [59]. In addition, we found that fragmentation was not due to an imbalance of anterograde and retrograde ER-Golgi transport. The main cause of this phenomenon may be the alteration of the levels of transport regulatory proteins, including Rab1, 2 and 8, and the SNARE protein syntaxin 5, the over-expression or depletion of these proteins reversing the damage.

The relationship between alterations in endosome-to-TGN and Golgi fragmentation in this pathology has not been extensively studied. However, it is clear that this route is altered in PD. Mutations of VPS35, a subunit of the retromer complex, which is involved in the recruitment of the WASH complex, has been related with autosomal-dominant late-onset PD and a point mutation was the causal agent in a Swiss family [89]. Mutations of Vps35 might indirectly affect the Golgi architecture because they can alter the trafficking of cathepsin D, which is involved in the degradation of α-synuclein, avoiding the formation of detergent-insoluble forms [90]. As indicated above, another protein important for this transport step is Rab7L1/Rab29, which regulates the retrograde transport of the mannose-6-phosphate receptor. This protein interacts with LRRK2, which is related to PD risk. Depletion of this GTPase induces the fragmentation of the TGN but not the cis/medial Golgi [91]. Therefore, it does not seem that alteration of endosome-TGN transport is a cause of GC fragmentation.

Alteration of the microtubule cytoskeleton has been observed in PD. LRRK2-induced neurodegeneration in Parkinsonian brains may be partly mediated by increased phosphorylation of beta tubulin damaging microtubule organization [92]. Parkin, an E3 ubiquitin ligase related with juvenile Parkinsonism, binds strongly to alpha/beta tubulin heterodimers and microtubules, stabilizing them [93]. Wild type but not Parkinson-associated forms of α-synuclein are able to bind microtubules and tubulin α2β2 tetramers, promoting microtubule nucleation and growth [94]. All of these factors may alter the stability of the microtubule, resulting in neurodegeneration. However, this does not seem to be the cause of Golgi fragmentation because it occurs before microtubule damage [59].

All in all, it seems that the most plausible cause of Golgi fragmentation in PD is alteration of the levels of some regulatory proteins, including Rabs and SNAREs, which results in the dysfunction of membrane transport at the early secretory pathway. ER stress, mitochondrial damage, altered calcium homeostasis and other causes may trigger the cells to change their expression.

## 7. Alzheimer’s Disease (AD)

The GCs of neurons affected by AD are reduced in size [95] and fragmented in a population of (3–6%) hippocampal neurons [96]. The pathological hallmarks of AD are the presence of intracellular neurofibrillary tangles and extracellular amyloid plaques in several brain regions. Extracellular amyloid plaques originate from toxic aggregates of β-amyloid peptides (Aβ), which are the result of the cleavage of the amyloid precursor protein (APP). The alteration of this proteolytic process induces the overproduction of toxic species [97,98,99,100]. This protein synthesized in the ER moves along the exocytic pathway to the plasma membrane and then enters the endocytic pathway. During its intracellular route, APP can be sequentially cleaved by three membrane-bound enzymes, known as α-, β-, and γ-secretases. Proteolysis of APP by α- and γ-secretases occurs in the plasma membrane and endosomes, respectively, and results in non-pathogenic fragments which have neurotrophic and neuroprotective properties. However, the proteolysis of APP by β- and γ-secretases in the acidic environment of the endosomes generates toxic Aβ [101]. Toxic peptides are released by exosomes or by other mechanisms into the extracellular space, where they are oligomerized to form protofibrils and fibrils [99]. Non-cleaved APP is transported back from the endosomes to the plasma membrane or to the TGN. Although the routes of APP and secretases are not fully understood, it is clear that the GC plays an important role in these processes. First, APP in steady-state is mostly located at the GC/TGN, where it is post-translationally modified in processes that include N- and O-glycosylation. BACE1, identified as brain-specific β-secretase, cycles between the GC/TGN and the plasma membrane and is mainly present in the TGN and endosomes in steady-state. It is synthesized as a precursor in the ER and requires extensive post-translational modifications (including N-glycosylation) and cleavage of the pro-peptide by a furin-like protease in the TGN. BACE1 is the rate-limiting enzyme in the production of toxic peptides and the physical interaction of this enzyme with APP in the same compartments is crucial to initiate the pathological process [102]. ADAM10 has been identified as the major α-secretase in brain neurons. It is also synthesized as a precursor in the ER and cleaved in the Golgi complex before being transported to the plasma membrane and then internalized. The transport of this enzyme to the synapse is also mediated by Golgi outposts, a process which is altered in AD [103]. The cleavage of APP by α- and β-secretase could be initiated at the TGN, after completion of O-glycosylation. The peptides generated by these enzymes are further processed by γ-secretase, which is a complex formed of four subunits. This complex is functionally assembled in the ER and some components, such as nicastrin, are glycosylated in the GC. Mutations in APP and the presenilins, PSN1 and PSN2, are responsible for early-onset AD (less than 10% of AD cases), whereas more genes (mainly involved in cholesterol metabolism, immune response and endocytosis) and environmental factors are involved in sporadic AD [104,105].

Based on studies in AD mice and tissue culture models, it has been proposed that the main cause of Golgi fragmentation in AD is the phosphorylation of GRASP65 by Cdk5 [60,101]. Thus, this mechanism could be similar to the ribbon unlinking and unstacking observed during mitosis, when phosphorylation of GRASP65 at different sites is necessary [28]. This study shows that Golgi fragmentation accelerated APP transport and increased the production of Aβ, whose accumulation triggered the activation of the kinase. Thus, it seems that there is a feedback mechanism between Golgi fragmentation and Aβ accumulation. However, it is not easy to explain how Golgi fragmentation can enhance transport. It has been demonstrated that completely unstacked Golgi are associated with a high number of vesicles and buds, which may explain the higher rate of transport [106]. However, the transport is not altered in cells where the ribbon is fragmented but there is no unstacking. The GC in AD transgenic mice showed much altered Golgi with swollen cisternae but it is not known if they are representative of the GC of neurons from AD patients. In these models, the authors did not observe degradation of Golgi structural proteins, which might also be responsible for Golgi fragmentation.

The alteration of endosome-to-TGN as a cause of Golgi fragmentation has not been studied in AD. However, there is evidence that the endosomal system is altered [107,108]. Neurons in the early stages of AD patients have enlarged endosomes in which APP accumulates. Moreover, some endocytosis-related genes have been associated with AD variants. Two of these genes, sortilin-related receptor and VPS35, are involved in retromer-mediated retrograde transport to the TGN of APP and BACE1 [109,110].

Another cause of Golgi fragmentation in AD may be the alteration of the microtubule network due to the formation of neurofibrillary tangles as a consequence of the hyperphosphorylation of tau protein by cdk5 (and other kinases). Tau is a microtubule-associated protein mainly located in axons and has an important role in microtubule assembly and stabilization [111]. Neurofibrillary tangles impair microtubule integrity and might block membrane transport. There is a link between tau secretion and Golgi dynamics, given that Golgi fragmentation induced by the suppression of the GTPase rab1A increases tau secretion [112]. In human patients, most of the hippocampal pyramidal neurons with neurofibrillar tangles have a highly altered GC in contrast with the low percentage of cells without it [113]. However, populations of neurons from AD patients without neurofibrillary tangles were seen to have fragmented GC, and in cells with such structures, the GC is deformed but not fragmented [114]. The GC was reduced in size in neurons from a neurofibrillary tangle P301S tauopathy mouse model [115]. Given that there was no Aβ pathology in this model, the authors deemed that GC damage was not a consequence of it. Moreover, alteration of the GC was observed during the initial steps of AD pathology, when plaques and tangles were scarce [58]. Thus, there seems to be no direct relationship between tangles and GC fragmentation. In fact, neurodegeneration can occur without tangles [116].

Taking all the above into account, it seems that the major cause of Golgi fragmentation in AD is phosphorylation of the stacking protein GRASP65. If this is the case, ribbon unlinking could be similar to the physiological fragmentation observed in mitosis, a process initiated by the activation of cdk5 by an unknown mechanism.

## 8. Amyotrophic Lateral Sclerosis (ALS)

ALS was the first neurodegenerative disease in which Golgi fragmentation was described [117,118]. This alteration was described in all forms of ALS in patients and animal models [119]. As described in other neurodegenerative diseases, Golgi fragmentation is an early event that occurs in a preclinical stage, prior to other pathological events such as activation of the injury transcription factor ATF3, neuromuscular denervation, axon retraction and activation of the mitochondrial apoptotic pathway [120].

One of the hallmark pathological features of ALS is the presence of protein aggregates in the degenerating lower motor neurons, such as Bunina bodies and several types of cytoplasmic inclusions of ubiquitinated proteins. The first are small eosinophilic inclusions containing transferrin, cystatin C, peripherin and the VPS (vacuolar protein sorting) 10 family [121]. The major component of ubiquitinated cytoplasmic inclusions in all types of ALS is TDP-43 (TAR DNA-binding protein of 43-kDa), which is a DNA and RNA binding protein predominantly located in the nucleus [122]. SOD1 (Cu/Zn superoxide dismutase-1) inclusions containing ubiquitinated misfolded SOD1 were observed in patients carrying mutations in this and other ALS-related genes [123]. The GC is fragmented in most motor neurons containing inclusions [119]. Intraneuronal inclusions do not contain Golgi proteins such as MG160 [40] or beta-COP [124]. Often, Bunina bodies and TDP-43 inclusions localize in the same neurons [121]. The fragmentation of the GC is directly related with abnormal levels of cytoplasmic TDP-43 in the motor neurons of ALS patients [125]. Indeed, in cells showing skein-like inclusions, a type of TDP-43 inclusion, the GC is not detected under the light microscope, indicating a high degree of fragmentation [125]. However, Golgi fragmentation precedes SOD1 inclusions [120].

Most cases of ALS are sporadic, but 10% are caused by mutations in different genes, mainly TDP-43, FUS (fused in sarcoma), optineurin, SOD1 and C9orf72 (hexanucleotide repeat expansion of the non-coding region of reading frame 72 on chromosome 9) [126,127]. Most of the ALS-related proteins are involved in trafficking [128,129]. Mutant forms of the RNA-processing proteins TDP-43 and FUS inhibit ER-to-ERGIC transport by preventing the incorporation of cargo into COPII vesicles. Mutant SOD1, instead, inhibits ERGIC-to-Golgi transport by destabilizing microtubules. The inhibition of COPII-mediated ER exit and ERGIC-to-Golgi transport may trigger Golgi fragmentation [128,130]. Both transport steps are Rab1-dependent, suggesting that this GTPase is a common target of these pathological events. Rab1 is recruited in motor neuron inclusions in patients with sporadic ALS and cellular models of ALS [130] and the consequence may be altered transport. Alteration of post-Golgi traffic may also be responsible for the Golgi fragmentation observed in this pathology. Thus, mutations in optineurin, an adaptor protein involved in myosin VI-dependent post-Golgi traffic, also induce Golgi fragmentation [131]. Mutant but not wild-type SOD1 interacts with chomogranins at the TGN and may alter the secretory pathway in some way. In fact, the mutated form of SOD1 enters this pathway by means of unknown mechanisms and is secreted extracellularly [132]. C9orf72 shows homology to a guanine nucleotide exchange factor (GEF), which activates Rab (and Rho) GTPases, including Rab 5, 7 and 11, which regulates endosomal trafficking [133].

Golgi fragmentation in ALS could also be due to microtubule alteration. Mutant SOD1 decreases the levels of acetylated tubulin, inducing less stable microtubules and affecting transport in the early secretory pathway [130]. However, as indicated above, TDP-43 and FUS mutants inhibit ER-Golgi transport by a microtubule-independent mechanism. TDP-43 interacts with different components of the neuron cytoskeleton, binding their RNAs [134]. Other ALS-related genes where the microtubule cytoskeleton may be dysfunctional and altered are TUBA4A (alpha-tubulin isofom), VAPB (vesicle-associated membrane protein B) and dynactin [127,128]. In neurons from mutant SOD1 mice, there is an up-regulation of stathmin 1, a microtubule-destabilizing protein, and the GC is fragmented in 30% of the neurons accumulating this protein [135]. The defective polymerization of Golgi-associated microtubules due to the progressive accumulation of stathmin 1/2 in mutant SOD1 mice models may trigger Golgi fragmentation [136]. In this model, fragmentation may be enhanced by dislocation of GM130 to cytosol or the interaction of mutant SOD1 with components of the secretory molecular machinery. However, early events in Golgi fragmentation are independent of stathmin 1 accumulation [135]. There is a TDP-43-depend down-regulation of the microtubule regulator stathmin 2 in the spinal cords of familiar ALS patients, but it does not occur in FUS and C9orf72 mutants [137]. Thus, the role of the members of the stathmin family in Golgi fragmentation may not be a general mechanism, so microtubule disorganization does not seem to be a primary cause of Golgi fragmentation. It has been proposed that Golgi fragmentation in ALS is due to a defective cross-talk between the microtubule cytoskeleton and COPI-coated vesicles [20,138]. In these studies, progressive motor neuronopathy mice have been used as models of motoneuron disease. As indicated above, in these mice lacks a Golgi-associated chaperone in the formation of microtubules [139]. The motor neurons of these animals also present Golgi fragmentation very early after birth, before any pathological symptoms appear. The authors proposed that the lack of microtubules impairs COPI assembly, resulting in fragmentation. Concomitantly, COPI can rescue the polymerization of microtubules and restore Golgi morphology. On the other hand, Golgi fragmentation is not observed in neurons from mice expressing human NF-H, which develop a motor neuropathy resembling ALS, despite the massive accumulation of neurofilaments in the soma [114].

Together, it seems that the major cause of Golgi fragmentation is due to disturbed Rab1-dependent transport in the early secretory pathway, which, as a consequence, produces an imbalance in the transport to and from the GC and also has dramatic effects in the cytoskeleton.

## 9. Lessons from Other Neurodegenerative Diseases

The GC in neurons from other neurodegenerative diseases has not been fully investigated but might help to obtain conclusions about causes of their alteration.

The Golgi appears fragmented in the ballooned neurons (neurons with distended, vacuolated soma) in patients with corticobasal degeneration and Creutzfeldt-Jakob disease [140]. In contrast to what happens in other neurodegenerative diseases, these Golgi elements remain in the cell center, whereas the periphery of the cytoplasm is occupied by phosphorylated neurofilaments (which are usually localized in the axon rather than soma). However, as indicated above, fragmentation is not a consequence of the massive accumulation of neurofilaments.

Spinal muscular atrophy (SMA) is a group of progressive neurodegenerative that affects lower motor neurons. The most frequent form is due to homozygous deletions in the SMN1 gene [141]. SMN (survival of motor neurons) proteins are ubiquitously expressed and have been involved in different functions, including snRNP transport, nuclear RNA splicing and profilin and actin-dependent axonal transport [142]. Interestingly, SMN proteins bind the coatomer subunit α-COP [143] and low levels of SMN proteins affect COPI-dependent ER-Golgi transport [144]. Fibroblasts from SMA patients have an altered Golgi, a morphology that can be rescued by over-expression of α-COP or SMN proteins [144]. Thus, Golgi alteration in SMA samples could be due to suboptimal α-COP functioning. This relationship remains to be demonstrated in neurons. Infrequent variants of SMA have been associated with mutations in the BICD2 (a dynein adaptor protein) and DYNC1H1 (dynein heavy chain) genes [141]. It is not known whether the GC of these patients is damaged, but if it is demonstrated that they are, alteration of microtubule-dependent transport might be the cause.

Huntington’s disease is caused by polyglutamine expansion in the N-terminal of the huntingtin protein. This protein is involved in post-Golgi transport along the microtubule and actin cytoskeleton. It is also required for ER-to-Golgi transport [145]. To the best of our knowledge, fragmentation of the GC has not been observed in patients with this disease. However, over-expression of mutant huntingtin protein increases the level of vesiculation of GC, especially the number of clathrin-coated vesicles [146]. One of the Golgi partners of this protein is optoneurin, which connects it with the actin-based motor myosin VI. Optoneurin, as described above, is one of the genes related with ALS.

The GC is also smaller and fragmented in neurons from the inferior olivary nucleus in patients with multiple system atrophy [74]. This phenomenon was observed in its early stages after central lesions. All neurons with α-synuclein cytoplasmic inclusions had fragmented GC but 56–86% of the cells with fragmented GC did not have inclusions.

The Golgi is also fragmented in cellular models of type 2 spinocerebellar ataxia, a disease caused by the expansion of polyglutamine repeats in ataxin-2 [147]. This protein is located in the GC and the mutant form breaks the normal morphology of this organelle.

## 10. Conclusions

Fragmentation of the GC may significantly alter neuronal physiology. Ribbon fragmentation alters polarized transport. Moreover, the GC in neurons is an important microtubule-organizing center and its alteration may significantly alter the cytoskeleton and associated activities. Thus, fragmentation may induce failures in transport to axons and synapses as well as dendrites. In PD, the specific consequence of Golgi fragmentation could be the reduced delivery of dopamine transporters in the synapsis, which may result in the limited uptake of this neurotransmitter into the vesicles and its accumulation in the cytosol. Oxidation of dopamine may result in the formation of toxic metabolites, which may modify α-synuclein, increasing its toxicity [87]. In AD, fragmentation may modify the processing of APP, leading to more toxic species. In ALS, it may also induce axon degeneration. The Golgi ribbon negatively regulated autophagy [42] so that its fragmentation may up-regulate this mechanism. Finally, Golgi alteration may trigger a stress response and, as consequence, may result in cell death. However, all the consequences of fragmentation are difficult to understand because there are numerous signaling molecules in the GC that are involved in a large number of signaling pathways, and alteration of the architecture of the GC may modulate signaling [148].

A summary of the main causes of Golgi fragmentation described above can be found in Table 1. After analysis of the causes of Golgi ribbon breakage in neurodegenerative diseases, a few conclusions are offered. It is clear that protein aggregation and the formation of aberrant filaments is not the cause of the fragmentation, although both can alter the distribution of the GC within the cell. Thus, for instance, fragmented GC enters dendrites in ALS neurons [120]. Microtubule alterations are probably not the primary cause of fragmentation, but rather an important early consequence. This information can be extrapolated from other neurological diseases. Thus, severely edematous neurons have fragmented GC but microtubules are intact [6].

Alteration of the intracellular transport, especially in the early secretory pathway, seems to be the main cause of Golgi fragmentation in neurodegenerative diseases including PD, AZ, ALS and SMA. As described above, the role of ER-to-Golgi Rab1-dependent transport in Golgi fragmentation has been demonstrated in PD [59,149] and ALS [130]. Rab1 protects against α-synuclein toxicity [82] and also partially corrects motor deficits in animal models of PD [149]. Rab1 is protective in ALS because the over-expression of this GTPase restores the macroautophagy inhibited by mutant FUS [150]. Golgi fragmentation induced by Rab1 suppression increases Tau secretion in neurons [112]. Although a similar role has not been demonstrated in AD, Rab1B regulates the processing of APP [151] and Rab1A enhances γ-secretase activity [152]. Curiously, Rab1 and SMA genes are physically linked [153]. The first role ascribed to this Rab protein was the regulation of ER-Golgi transport through recruitment of the tethering factor p115 and v-SNAREs in ER-derived COPII-coated vesicles, but also the formation of the Golgi ribbon by binding Golgin-84 [154]. Thus, this GTPase is a Golgi housekeeping protein. More recently, it has also been seen to be involved in signaling pathways regulating cell migration, the metabolism of nutrients, cell proliferation, cytokinesis and autophagy. Rab1 is implicated in cancer, cardiomyopathy, infectious diseases as well as neurodegenerative diseases. It seems plausible that Rab1 has a key role in Golgi fragmentation but also in other pathological processes during the early stages of neurodegeneration.

More studies are necessary to understand the mechanism that fractures the Golgi morphology and its consequences. For instance, more ultrastructural studies of the Golgi architecture are necessary in human samples. It is necessary to ascertain the degree of ribbon fragmentation, cisternae unstacking and vesiculation because the molecular machineries involved in these processes are not the same. Only 3D analysis of the GC by powerful imaging techniques, such as electron tomography, can give a complete picture of the alterations of the architecture of this organelle during the progress of the disease. One of the main questions is why only a small group of neurons is affected in these disorders even though the molecular machinery affected, for instance Rab1, is important in all cell types. Single cell multi-omics methodologies will help to solve this mystery. It would also be interesting to explore the role of glial cells in neurodegeneration in general and in the neuronal Golgi structure in particular. The question concerning a possible common cause for Golgi fragmentation in neurodegenerative diseases remains unsolved.

## Figures and Tables

**Table 1 cells-08-00748-t001:** Summary of the putative causes of Golgi fragmentation in neurodegenerative diseases (see text for details).

Neurodegenerative Disease	Cause of Fragmentation	Main Model Used	References
PD	**Alterations in the levels of Rab1:** 2 and 8 and the SNARE protein syntaxin 5	Differentiated PC12 cells treated with 6-hydroxydopamine or methamphetamine	[59]
	Prefibrillar aggregates of α-synuclein	COS-7 cells overexpressing α-synuclein	[75]
	Defects in Rab1 dependent ER-Golgi transport	Substancia nigra from rats injected with vectors for α-synuclein and Rab1	[149]
	Enhanced phosphorylation of Rab7L1 by mutant LRKK2 (TGN fragmentation)	HEK293 cells expressing Rab7L1 and LRRK2 mutant	[78]
AD	Accumulation of Aβ which causes activation of cdk5 and phosphorylation of GRASP65	Transgenic mice/CHO cells expressing APP and PS1 mutants	[60]
	Accumulation of phospho-tau	Neurons from neocortex and hippocamous of AD patients	[113]
	Tau secretion	Primary cortical neurons	[112]
ALS	Endosomal abnormalities due to altered dynein-dependent transport	SOD1-ALS mouse	[120]
	Mutations in optineurin	Motor neuron-like NSC-34 cells	[131]
	Interruption of ARF1/TBCE cross-talk that coordinate COPI formation and microtubule polymerization	Motor neurons from progressive motor neuropathy mice	[20]
	Rab1-dependent ER-Golgi transport	Neuro2a cells transfected with TDP-43, FUS or SOD1	[130]
	Stathmin 1/2-triggered microtubule destabilization	Transgenic SOD1mutants mouse motor neurons	[136]
SMA	Low levels of α-COP	SMA patient fibroblasts	[144]

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
