# Peer review of "Golgi Fragmentation in Neurodegenerative Diseases: Is There a Common Cause?"

_cells, 2019, doi:10.3390/cells8070748_

Round 1
Reviewer 1 Report
This manuscript summarized the current findings and revealed that Golgi fragmentation causes the neurodegenerative diseases. This review article focused on the key question and clearly described the previous studies.
Here are a few additional minor suggestions.
1 Briefly, Golgi morphology is controlled by three types of protein: (1) microtubules and their associated motor proteins; (2) Golgi matrix proteins, including GRASP55, GASP65, GM130, Golgin-45, Golgin-84 and Golgin-160; (3) the proteins of the Golgi transport machinery. The authors should add this information to strengthen the rationale.
2 Golgi disruptions impair accurate glycosylation and sorting. Furthermore, golgi fragmentation leads to cell death. The authors should classify it to explain the reasons more clearly.
3 The authors should use the same font size in the main text.
4 It would be better to discuss how powerful molecular, structural, genetic and imaging tools to extend what is now known.
Author Response
Comments to Reviewer 1
This manuscript summarized the current findings and revealed that Golgi fragmentation causes the neurodegenerative diseases. This review article focused on the key question and clearly described the previous studies.
Thank you for your helpful comments
Here are a few additional minor suggestions.
1 Briefly, Golgi morphology is controlled by three types of protein: (1) microtubules and their associated motor proteins; (2) Golgi matrix proteins, including GRASP55, GASP65, GM130, Golgin-45, Golgin-84 and Golgin-160; (3) the proteins of the Golgi transport machinery. The authors should add this information to strengthen the rationale.
The information in the text has been completed with these proteins
2 Golgi disruptions impair accurate glycosylation and sorting. Furthermore, golgi fragmentation leads to cell death. The authors should classify it to explain the reasons more clearly.
We do not think that cell death is triggered by altered glycosylation or sorting. In fact, fragmentation of the ribbon neither alter glycosylation nor transport rate, although there is an alteration of polarized sorting (reference 16). As indicated in the text, it has been postulated that fragmentation triggers Golgi stress response and, at the end, induce cell death (reference 35).
3 The authors should use the same font size in the main text.
Font size and type has been checked
4 It would be better to discuss how powerful molecular, structural, genetic and imaging tools to extend what is now known.
Mention to some of these methods such as electron tomography and single cell multi-omics have been included in the new version (lines 554-561)

Reviewer 2 Report
Review cells-555610
Martinez-Menarguez present a review manuscript about the role of Golgi fragmentation in neurodegenerative diseases. They review the literature about Golgi structure and organization in general and in neurons. This is followed by a description of the fragmentation of the Golgi as typical for many neurodegenerative diseases and finally the authors review what is know about Golgi fragmentation in more detail in selected diseases like ALS, PD and AD.
The topic is highly interesting and I did not find a recent review similar to the one presented here. There seem to be a number of recent reviews about Golgi fragmentation in selected diseases, but none covered the general aspects in neurodegeneration. A review on the topic is therefore important. However, before recommending it for publication I think some revision would improve the manuscript. Overall I find it a bit, sorry, dry and boring and I suggest the authors should make it a bit more interesting. Maybe simply by adding one or more figures to illustrate the main points and maybe by a table listing the pros and cons for GC fragmentation being cause or consequence in neurodegeneration.
There are also many typographical and grammar errors in the text, words missing etc. I´ll list some of them, but the authors need to go through the text more thoroughly, ideally involving a native speaker.
My specific comments:
line 14: there is a superfluous “a”
19: “to” instead of “with”, “try to search” instead of “trying”.
33: what are „sensible genes“?
36: verb missing, probably „affected“
40: delete the „s“ from others
44: I suggest to add 1-2 sentences what the review is about, how it is structured etc. Otherwise the next chapter „Golgi Ribbon“ comes a bit sudden. Something like „In this review we will first give an overview,....
46: mention that dictyosomes is the term for Golgi in plants (as far as I know).
70: I would make more clear that the Golgi is an additional site for nucleation, next to the MTOC. On first reading I had the impression you want to say it is the only site.
102ff: Ref. missing for the GRASP findings.
149: Outposts misspelled
167: “a” in front of phenomenon missing
177: did not show
183: A sentence with a conclusion for this paragraph would be nice.
196: I don´t understand the argument that the distance is very large betweenERES, GC, etc. in neurons. I would have thought the opposite, given that the cell body is so small.
205: delete the “be”.
214: I would first explain that there is reversible GC fragmentation, now it comes out of the blue.
Maybe for this paragraph a figure or scheme would be nice to illustrate the potential causes of GC fragmentation.
222: in, not into
225: delete “of the”
238ff: You would have to postulate here that there must be reversible fragmentation, because you say the neurons with LBs (i.e. in a later stage) show no or very rarely GC fragmentation. This should be discussed.
299ff: Can you specify the regulatory proteins that are altered? In this paragraph you mention mainly Rabs, but for me these are more basic transport proteins, not regulatory. Maybe you can define what you mean.
305ff: Can you specify that a bit more? What means affected? Is the fragmentation something common or rare like you state for the LB neurons?
316: Most abeta will be released conventionally via the secretory pathway, the exosome route is certainly a minor route.
335: I don´t think presenilins are glycosylated. If I´m wrong please provide original reference(s).
340: What about this mechanism in other diseases? Wouldn´t that be interesting to discuss? Maybe this is the common cause?
376: add “in AD” after Golgi fragmentation.
420: As indicated above. Where? I did not find that.
422: this is not true. What is not true? Unclear what this refers to
430: “did not reject” I think is not a valid expression
438/39: several typos
447: is due to disturbed Rab1…
487: summary/conclusion?
489: ribbon fragmentation alter polarized transport. Maybe I missed that but did you mention that before? If not please cite a reference.
521: two times been
Author Response
Martinez-Menarguez present a review manuscript about the role of Golgi fragmentation in neurodegenerative diseases. They review the literature about Golgi structure and organization in general and in neurons. This is followed by a description of the fragmentation of the Golgi as typical for many neurodegenerative diseases and finally the authors review what is know about Golgi fragmentation in more detail in selected diseases like ALS, PD and AD.
The topic is highly interesting and I did not find a recent review similar to the one presented here. There seem to be a number of recent reviews about Golgi fragmentation in selected diseases, but none covered the general aspects in neurodegeneration. A review on the topic is therefore important. However, before recommending it for publication I think some revision would improve the manuscript. Overall I find it a bit, sorry, dry and boring and I suggest the authors should make it a bit more interesting. Maybe simply by adding one or more figures to illustrate the main points and maybe by a table listing the pros and cons for GC fragmentation being cause or consequence in neurodegeneration.
Thank you very much for your helpful comments. A table with the main theories about the causes Golgi fragmentation has been included in the new version of the manuscript. We agree that it can help to understand the this complex process. During the elaboration of this review we thought about the possibility of to add a figure but we conclude that it could be very complex because we are talking about several diseases and many different transport step/machineries.
There are also many typographical and grammar errors in the text, words missing etc. I´ll list some of them, but the authors need to go through the text more thoroughly, ideally involving a native speaker.
The manuscript has been reviewed by MPDI English editing
My specific comments:
line 14: there is a superfluous “a”
Ok
19: “to” instead of “with”, “try to search” instead of “trying”.
Ok
33: what are „sensible genes“?
This word has been removed
36: verb missing, probably „affected“
Ok
40: delete the „s“ from others
Ok
44: I suggest to add 1-2 sentences what the review is about, how it is structured etc. Otherwise the next chapter „Golgi Ribbon“ comes a bit sudden. Something like „In this review we will first give an overview,....
We completely agreed with it and a new sentence has been included (lines43-46)
46: mention that dictyosomes is the term for Golgi in plants (as far as I know).
This old term is synonymous of Golgi stack as you can see, for instance, in the famous taxt book of Cell Biology Alberts’s. In Spain, we use it very commonly (the translation of “stack” do not sound very well in Spanish!!!) However, it is true that nowadays is used specially for plants.
70: I would make more clear that the Golgi is an additional site for nucleation, next to the MTOC. On first reading I had the impression you want to say it is the only site.
This sentence has been clarified (lines 77-80)
102ff: Ref. missing for the GRASP findings.
A reference has been included
149: Outposts misspelled
Ok
167: “a” in front of phenomenon missing
Ok
177: did not show
Ok
183: A sentence with a conclusion for this paragraph would be nice.
A conclusion sentence has been included (lines 194-198)
196: I don´t understand the argument that the distance is very large betweenERES, GC, etc. in neurons. I would have thought the opposite, given that the cell body is so small.
The large distance between the GC and the plasma membrane (for instance, the axon ends) in neuron is evident. The distance between some ERES (those associated with dendrites) and the somatic Golgi complex has been also demonstrated (reference 51). This reference has been included in the new version of the manuscript
205: delete the “be”.
214: I would first explain that there is reversible GC fragmentation, now it comes out of the blue. Maybe for this paragraph a figure or scheme would be nice to illustrate the potential causes of GC fragmentation.
We do not agree with the reviewer. We think that the idea is very simple and do not need extra support.
222: in, not into
Ok
225: delete “of the”
Ok
238ff: You would have to postulate here that there must be reversible fragmentation, because you say the neurons with LBs (i.e. in a later stage) show no or very rarely GC fragmentation. This should be discussed.
This idea has been discussed in the new version (lines 253-256)
299ff: Can you specify the regulatory proteins that are altered? In this paragraph you mention mainly Rabs, but for me these are more basic transport proteins, not regulatory. Maybe you can define what you mean.
We include RAbs and SNARe proteins. Of course, there are more proteins affected in PD but it seems that those are directly related with Golgi fragmentation
305ff: Can you specify that a bit more? What means affected? Is the fragmentation something common or rare like you state for the LB neurons?
This point has been clarified. In fact, fragmentation is only observed in a small population of neurons
316: Most abeta will be released conventionally via the secretory pathway, the exosome route is certainly a minor route.
It has been described that the release of pathogenic Ab peptides occurs via exosomes (reference 99 and references there in). If pathogenic cleavage of APP by beta-secretase occurs in endosomes, their release will occur by exosomes. Anyway we added “exosomes and other mechanisms”
335: I don´t think presenilins are glycosylated. If I´m wrong please provide original reference(s).
You are right. Presenilins influence glycosylation but they don’t have glycosylation sites. This sentence has been corrected.
340: What about this mechanism in other diseases? Wouldn´t that be interesting to discuss? Maybe this is the common cause?
The mechanisms of other disease (more specifically SMA, because there are very few data in other diseases) has been discussed.
376: add “in AD” after Golgi fragmentation.
Ok
420: As indicated above. Where? I did not find that.
It was mention that mutant SOD1 destabilizes microtubules. Anyway this expression was removed to avoid confusion.
422: this is not true. What is not true? Unclear what this refers to
This sentence has been clarify
430: “did not reject” I think is not a valid expression
This expression has been removed
438/39: several typos
Ok
447: is due to disturbed Rab1…
Ok
487: summary/conclusion?
Conclusions
489: ribbon fragmentation alter polarized transport. Maybe I missed that but did you mention that before? If not please cite a reference.
It was already mentioned (lines 89-90). This is an important point because this important result from this study (reference 16) is usually non mentioned.
521: two times been
Ok
